# A Network Pharmacology and Multi-Omics Combination Approach to Reveal the Effect of Strontium on Ca^2+^ Metabolism in Bovine Rumen Epithelial Cells

**DOI:** 10.3390/ijms24119383

**Published:** 2023-05-27

**Authors:** Panpan Tan, Chenxu Zhao, Yong Dong, Zixin Zhang, Linshan Mei, Yezi Kong, Fangyuan Zeng, Yongqiang Wen, Baoyu Zhao, Jianguo Wang

**Affiliations:** College of Veterinary Medicine, Northwest A&F University, Yangling 712100, China

**Keywords:** strontium, Ca^2+^ metabolism, rumen epithelial cells, transcriptomics, proteomics, network pharmacology

## Abstract

Strontium (Sr) belongs to the same group in the periodic table as calcium (Ca). Sr level can serve as an index of rumen Ca absorption capacity; however, the effects of Sr on Ca^2+^ metabolism are unclear. This study aims to investigate the effect of Sr on Ca^2+^ metabolism in bovine rumen epithelial cells. The bovine rumen epithelial cells were isolated from the rumen of newborn Holstein male calves (*n* = 3, 1 day old, 38.0 ± 2.8 kg, fasting). The half maximal inhibitory concentration (IC50) of Sr-treated bovine rumen epithelial cells and cell cycle were used to establish the Sr treatment model. Transcriptomics, proteomics, and network pharmacology were conducted to investigate the core targets of Sr-mediated regulation of Ca^2+^ metabolism in bovine rumen epithelial cells. The data of transcriptomics and proteomics were analyzed using bioinformatic analysis (Gene Ontology and Kyoto Encyclopedia of genes/protein). Quantitative data were analyzed using one-way ANOVA in GraphPad Prism 8.4.3 and the Shapiro–Wilk test was used for the normality test. Results presented that the IC50 of Sr treatment bovine rumen epithelial cells for 24 h was 43.21 mmol/L, and Sr increased intracellular Ca^2+^ levels. Multi-omics results demonstrated the differential expression of 770 mRNAs and 2436 proteins after Sr treatment; network pharmacology and reverse transcriptase polymerase chain reaction (RT-PCR) revealed Adenosylhomocysteine hydrolase-like protein 2 *(AHCYL2)*, Semaphoring 3A *(SEMA3A)*, Parathyroid hormone-related protein *(PTHLH)*, Transforming growth factor β*2* (*TGF-β2)*, and Cholesterol side-chain cleavage enzyme *(CYP11A1)* as potential targets for Sr-mediated Ca^2+^ metabolism regulation. Together these results will improve the current comprehension of the regulatory effect of Sr on Ca^2+^ metabolism and pave a theoretical basis for Sr application in bovine hypocalcemia.

## 1. Introduction

Sr, as an alkaline-earth metal, is one of the essential trace elements in the body, which plays a key role in osteoporosis treatment and prevention [1,2]. Sr belonging to the same group in the periodic table as Ca exhibits similar physicochemical characteristics but distinct bone-seeking properties [2,3,4]. Numerous studies have suggested that Sr increases bone mineral density by preventing osteoclast activation and promoting osteoblast differentiation [3,5,6]. The team’s previous research found that Sr promoted proliferation and inhibited differentiation of bovine primary chondrocytes via the *TGFβ*/*SMAD* pathway [7]. Sr plays different roles in the presence of various concentrations of Ca^2+^; for instance, Sr inhibits bone regeneration at low Ca^2+^ concentration but enhances bone regeneration under high Ca^2+^ concentration [8]. A previous study showed that a Ca-free medium was more efficient for bovine oocyte activation with Sr [9]. While Sr is an agonist of the calcium-sensing receptor (*CaSR*) and has a lower affinity than Ca^2+^ [10], whether it affects Ca^2+^ homeostasis is yet unclear. 

At the onset of lactation, many cows are unable to adapt efficiently to the sudden increase in Ca demand, leading to blood Ca concentration decreases and causing hypocalcemia, thereby reducing cow performance and increasing the risk of other health disorders, such as metritis and ketosis [11,12,13,14]. In ruminants, Ca^2+^ homeostasis is mainly dependent on the regulation of intestinal Ca absorption, bone Ca resorption, and renal Ca reabsorption [15,16]. Ca absorption from the alimentary tract occurs in the small intestine as well as in the rumen [17]. The rumen and intestinal Ca absorption pathways are the major routes for obtaining Ca in vitro [18]. Given its chemical similarity with Ca, Sr has a similar transport and distribution pathway as Ca in the body; Sr can replace Ca in some physiological processes such as muscle contraction, blood clotting, and secretion of certain hormones [19,20]. Sr has been used as a Ca marker to measure intestinal Ca absorption [21,22]. Sr concentration in the blood plasma after an oral dose of strontium chloride (SrCl_2_) into the rumen can serve as an index of rumen Ca absorption capacity under different states of Ca homeostasis in sheep and dairy cows [19,23]. However, no study has investigated the effect of Sr on Ca absorption in the bovine rumen. 

To explore the effect of Sr on Ca^2+^ metabolism in bovine rumen epithelial cells, a model of Sr treatment in bovine rumen epithelial cells was established in the present study. RNA-sequencing-based transcriptomic profiles and DIA-based proteomic profiles were used to analyze the effect of Sr on bovine rumen epithelial cells in vitro, and network pharmacology was explored to investigate its effect on Ca^2+^ metabolism in bovine rumen epithelial cells.

## 2. Results

### 2.1. Effect of Sr on Viability of Rumen Epithelial Cells 

Rumen epithelial cells were successfully dissociated (Appendix A) and identified by the positive expression of cytokeratin 18 (CK18) and E-cadherin (Appendix A). The Lactate dehydrogenase (LDH) activity of cell culture supernatant was the highest on day 5, gradually decreased after day 6, and did not significantly change on day 7 (Appendix A). This trend in the change of LDH activity correlated with the growth of rumen epithelial cells. 

After treatment with different doses of Sr for 24 h, the IC50 value of rumen epithelial cells was 43.21 mmol/L (Figure 1A). Cell cycle analysis results showed that the proliferation index (PI) value was significantly increased in the 20 mmol/L doses of Sr compared to the 0 mmol/L Sr group (*p* < 0.01) (Figure 1B,C). Hence, the 0, 1, 10, and 20 mmol/L Sr concentrations were used for subsequent experiments. 

### 2.2. Effect of Sr on Intracellular Ca^2+^ Level in Rumen Epithelial Cells 

The Ca^2+^ staining fluorescence value was significantly increased in the 20 mmol/L Sr treatment group compared to the 0 mmol/L Sr group (*p* < 0.05) (Figure 1D), suggesting that the intracellular Ca^2+^ level had an increasing trend with increasing Sr doses. 

### 2.3. Analysis of Differentially Expressed Genes (DEGs)

After treatment with Sr, the DEGs in rumen epithelial cells were visualized using volcano plots and hierarchical clustering. In total, 770 DEGs comprising 446 upregulated and 324 downregulated genes were recorded (Figure 2A,B).

The DEGs were analyzed by Gene Ontology (GO) functional enrichment; a total of 235 GO terms were found to be significantly enriched (*q*-value < 0.05). The top 20 GO terms were all biological processes, such as developmental process, anatomical structure development, tissue development, anatomical structure morphogenesis, multicellular organism development, system development, and cell differentiation (Figure 2C). Kyoto Encyclopedia of Genes and Genomes database (KEGG) pathway analysis of DEGs showed that the most enriched pathways were those involved in pathways in cancer, amino sugar and nucleotide sugar metabolism, and nitrogen metabolism (*q*-value < 0.05) (Figure 2D). Protein-protein interaction network (PPI) analysis of the DEGs showed that a total of 130 core targets as selected by STRING and Cytoscape software (version 3.8.2) based on “betweenness”, “closeness”, and “degree” (Figure 2E).

### 2.4. Analysis of Differentially Expressed Proteins (DEPs)

The DEPs of rumen epithelial cells were visualized using volcano plots and hierarchical clustering, and 2436 DEPs were observed, including 1846 upregulated and 590 downregulated proteins (Figure 3A,B).

GO analysis showed that the DEPs in the Sr-treated and untreated cells were significantly enriched in 245 GO terms (*q*-value < 0.05). The top 20 GO terms mainly involved cellular components, including cytosol, cytoplasm, proteasome complex, cytoplasmic part, proteasome core complex, exocyst, proteasome accessory complex, and cell-substrate junction (Figure 3C). KEGG pathway analysis showed a total of 631 significantly enriched pathways (*q*-value < 0.05). The top 20 pathways included proteasome, purine metabolism, biosynthesis of amino acids, cysteine and methionine metabolism, amino sugar and nucleotide sugar metabolism, glycolysis/gluconeogenesis, fructose and mannose metabolism, metabolic pathways, alanine, aspartate and glutamate metabolism, and necroptosis (*q*-value < 0.05) (Figure 3D). The PPI of the DEPs showed a total of 400 core targets selected by STRING and Cytoscape software (version 3.8.2) based on “betweenness”, “closeness”, and “degree” (Figure 3E).

### 2.5. Association Analysis of the Transcriptome and Proteome

A total of 665 DEGs and 725 DEPs were displayed, including 69 molecules that were differentially expressed both at mRNA and protein levels, such as *PTHLH*, *CA2*, *TGFBI*, and *WWC2* (Figure 4A). The number of proteins and genes enriched was the highest in quadrant 5, followed by quadrants 6, 4, 2, and 8. The proteins and genes enriched in quadrant 5 were commonly expressed without any differences. The proteins and genes enriched in quadrants 4 and 6 might be associated with post-transcriptional or translation-level regulation. DEGs and DEPs in quadrants 3 and 7 showed similar expression patterns, which might be related to the genes that were not regulated or less regulated at the level of translation after transcription. A small number of proteins and genes showed lower abundance in quadrants 1 and 9. The association analysis of transcriptome and proteome data revealed a Pearson’s correlation coefficient of 0.1869 (*p*-value = 0). These results explained that the abundance of most DEPs did not correlate with the corresponding transcriptional levels (Figure 4B). 

The DEGs and DEPs were analyzed by GO functional enrichment, and a total of 24 GO terms were significantly enriched (*q*-value < 0.05). The top 20 GO terms mainly involved biological processes such as nucleotide-sugar biosynthetic process, UDP-N-acetylglucosamine biosynthetic process, amino sugar biosynthetic process, UDP-N-acetylglucosamine metabolic process, nucleotide-sugar metabolic process, and amino sugar metabolic process (Figure 4C). KEGG pathway analysis showed that the DEGs and DEPs were significantly enriched in three pathways (*q*-value < 0.05), including amino sugar and nucleotide sugar metabolism, metabolic pathways, and fructose and mannose metabolism (Figure 4D).

### 2.6. Networks and Enriched Functions in Ca^2+^ Metabolism-Associated Genes

A total of 1143 Ca^2+^ metabolism-associated genes were identified, and 9 common targets were identified between hypocalcemia, transcriptome, and proteome profiles (Figure 5A). These common targets were considered as core targets of Sr action on Ca^2+^ metabolism. Among these targets, the expression levels of *AHCYL2*, WW-and-C2-domain-containing protein 2 (*WWC2*), *SEMA3A*, *PTHLH*, and Carbonic anhydrase II (*CA2*) were significantly upregulated, and *CYP11A1*, Uroplakin 1b (*UPK1B*), *TGF-β2*, and Transforming growth factor, β-induced (*TGFBI*) were significantly downregulated in transcriptomics and proteomics (Figure 5B). These core targets showed a positive correlation in transcriptomics and proteomics (R^2^ = 0.9420, *p* < 0.01) (Figure 5C).

### 2.7. RT-PCR Analysis of the Effect of Sr on Targets Changes

As shown in Figure 5E, the mRNA expression levels of *AHCYL2*, *WWC2*, *SEMA3A*, *PTHLH*, and *TGFBI* significantly increased with the Sr concentration increase (*p* < 0.05, *p* < 0.01) at 10 and 20 mmol/L Sr doses, and the expression of *UPK1B* significantly increased at 1 and 10 mmol/L Sr doses (*p* < 0.01). The *CYP11A1* expression was significantly upregulated at 1 mmol/L Sr doses (*p* < 0.05) and, in contrast, significantly decreased at 20 mmol/L Sr doses (*p* < 0.05). The *TGF-β2* mRNA expression level was significantly downregulated (*p* < 0.01) at 10 and 20 mmol/L Sr doses (*p* < 0.01), while the *CA2* expression level did not change significantly after Sr treatment. The RT-PCR results of *CYP11A1*, *AHCYL2*, *WWC2*, *SEMA3A*, *PTHLH*, and *TGF-β2* were consistent with transcriptomics results. Transcriptome results of *CYP11A1*, *AHCYL2*, *SEMA3A*, *PTHLH,* and *TGF-β2* were positively correlated with RT-PCR results (R^2^ = 0.7785, *p* < 0.05) (Figure 5D). The normality test results of *CYP11A1*, *AHCYL2*, *UPK1B*, *WWC2*, *TGF-β2*, *SEMA3A*, *PTHLH, CA2*, and *TGFBI* were not significantly changed (*p* > 0.05), and were normally distributed (Appendix A). These results proved that the core targets identified by association analysis between network pharmacology, transcriptomics, and proteomics were reliable.

## 3. Discussion

Hypocalcemia is a metabolic disease caused by the homeostatic imbalance of blood Ca^2+^ concentration in cows, which impacts their health, future milk production, and reproductive performance [24,25]. Research suggests that blood plasma Sr level can be used as an index of rumen Ca absorption capacity in sheep and dairy cows [19,23]. A few studies have shown that Sr causes intracellular Ca^2+^ concentration oscillations generation in rats and mice [26,27]. In the present study, a model of Sr treatment bovine rumen epithelial cells was established in vitro. The intracellular Ca^2+^ concentration significantly increased in bovine rumen epithelial cells treated with 20 mmol/L Sr. To further explore the effect of Sr on Ca^2+^ homeostasis in rumen epithelial cells, the characteristics and differences of each group were comprehensively analyzed using RNA-sequencing-based transcriptomics and DIA-based proteomics. Additionally, 770 DEGs and 2436 DEPs were found in an analysis of differential expression between the control and 20 mmol/L Sr group, including 69 differential expressions molecular both in gene and protein levels; these overlapping DEGs/DEPs were mainly related to the cellular metabolism. Furthermore, the number of DEPs was more than the number of DEGs in quadrants 4 and 6, and the association between transcriptome and proteome was weak, this result suggests that the effect of Sr on rumen epithelial cells might be regulated by post-transcriptional modifications [28].

In ruminants, the maintenance of blood Ca^2+^ concentration mainly relies on the regulation of intestinal Ca absorption, bone Ca resorption, and renal Ca reabsorption [29]. Intestinal Ca absorption is a major pathway for external Ca intake [18]. Sr and Ca have the same mechanisms for absorption from the gastrointestinal tract and bone accumulation in the human body [29]. Stimulation of Ca absorption via transcellular transport gains can counteract hypocalcemia at the onset of lactation. Ca absorption and transport are mainly active and transcellular transport in the rumen. Transcellular transport in the epithelium is regulated by calbindin-D9K, transient receptor potential cation channel subfamily V member 5 (TRPV5), TRPV6, plasma membrane Na^+^/Ca^2+^ exchanger (NCX1), and plasma membrane Ca^2+^ ATPase 1b (PMCA1b). The vitamin D metabolite, 1,25-dihydroxy vitamin D3 [1,25-(OH)_2_D_3_], parathyroid hormone (PTH), and fibroblast growth factor 23 (FGF23) are prominent hormones controlling the Ca^2+^ balance [30,31]. In the rumen of goat, sheep, and bovine, the calbindin-D9K, TRPV5, and TRPV6 expression levels are weak or have no expression [18,32,33]. To further screen the core targets of Sr-mediated regulation of Ca^2+^ metabolism, in the present study, the combination of network pharmacology and multi-omics have obtained 9 cores targets as follows: *PTHLH*, *SEMA3A*, *TGF-β2*, *TGFBI*, *CA2*, *CYP11A1*, *WWC2*, *UPK1B*, and *AHCYL2*.

*PTHLH* (also called the *PTHrP*) is a key component in Ca^2+^ metabolism during pregnancy [34]. *PTHLH* increases chondrocyte sensitivity to 1,25(OH)_2_D_3_ by enhancing vitamin D receptor (*VDR*) production [35,36]. The change in *PTHrP* expression level was correlated with the changes in Ca^2+^ concentration in goat mammary epithelial cells and serum [37,38]. *SEMA3A* is a secreted glycoprotein that functions as a potent osteoprotective factor by synchronously inhibiting bone resorption and promoting bone formation [39]. *SEMA3A* treatment induces Ca^2+^ elevation in neurons [40,41]. *TGF-β2* plays a vital role in maintaining the homeostasis of cartilage tissue and regulating chondrocyte proliferation, differentiation, and apoptosis [42]. The presence of *TGF-β2* stimulates *FGF23* expression and induces cellular Orai1-mediated calcium influx from extracellular space in UMR106 cells [42]. *TGFBI* (also known as *βig-h3*) is a secretory extracellular matrix protein induced by *TGF-β* [43], the reduction in the level of *TGFBI* indirectly increased the concentration of extracellular Ca^2+^ [44]. *CA2* acts as a mediator of hormones that stimulate bone resorption and osteoclast differentiation [45]. Calcitonin increased the *CA2* activity and *PTH* had opposite effects in the human erythrocyte [46]. The expression level of *CA2* was stimulated by *PTH* and *1,25-(OH)_2_D_3_* [47,48]. *CYP11A1*, also known as cytochrome P450scc, is a member of the cytochrome P450 family of heme-containing enzymes that plays an important role in steroidogenesis [49]. *CYP11A1* activates vitamin D3 to produce noncalcemic products, such as 20(OH)D_3_ [50]. Intracellular Ca^2+^ levels has negatively correlated with *CYP11A1* [51]. There is no published literature on how *WWC2*, *UPK1B*, and *AHCYL2* regulate Ca^2+^ metabolism. In this study, *PTHLH*, *SEMA3A*, *CYP11A1*, and *TGF-β2* expression was consistent in RT-PCR and transcriptomics, the *UPK1B*, *CA2*, and *TGFBI* expression levels are incompatible with transcriptomics, probably owing to the interference of duplication on the quantitative results of sequencing. The differences in *AHCYL2*, *SEMA3A*, *PTHLH*, *TGF-β2*, and *CYP11A1* gene expression levels and coefficients between the RNA-sequencing and RT-PCR results may reflect the underlying targets of Sr-mediated Ca^2+^ metabolism regulation. The molecular mechanisms of these molecules mediated by Sr in Ca^2+^ metabolism regulation require further investigation.

In conclusion, the RNA-sequencing-based transcriptomic profiles and DIA-based proteomics profiles of Sr-treated bovine rumen epithelial cells revealed 770 DEGs and 2436 DEPs. The most highly expressed genes and proteins were involved in metabolism. The combined network pharmacology analysis and RT-PCR validation revealed 5 core targets that were potentially involved in the Sr-mediated Ca^2+^ metabolism regulation, namely, *AHCYL2*, *SEMA3A*, *PTHLH*, *TGF-β2,* and *CYP11A1*. The results of this study will help to understand the regulatory effect of Sr on Ca^2+^ metabolism and provide a theoretical basis for Sr application in bovine hypocalcemia.

## 4. Materials and Methods

### 4.1. Isolation and Culture of Primary Bovine Rumen Epithelial Cells

The rumen tissue was collected from newborn Holstein male calves (*n* = 3, 38.0 ± 2.8 kg body weight) within 15 min after euthanasia, serial trypsin digestions were used to isolate primary bovine ruminal epithelial cells as previously described [52,53]. The collected rumen tissue was washed several times with ice-cold 0.9% (*w/v*) sodium chloride (NaCl, pH 7.0) until no visible rumen contents remained. The rumen epithelium was bluntly dissected, washed twice with phosphate-buffered saline (PBS) containing penicillin (2500 U/mL) and streptomycin (2500 mg/mL) for 30 min at 37 °C, and then washed with PBS containing amphotericin B (1000 U/mL) and gentamicin (12 µg/mL) for 30 min at 37 °C. The rumen epithelium was aseptically cut into small pieces (3–4 cm^2^), washed with PBS, and subjected to serial trypsinization (Sigma, St. Louis, MO, USA) with trypsin-ethylenediaminetetraacetic acid (EDTA) solution (0.25% trypsin and 0.02% EDTA-Na_2_) at 37 °C. The trypsin-EDTA solution was freshly replaced every 8 min; the first fraction of the supernatant was discarded, and the subsequent four fractions were separately strained through 50-mesh and centrifuged for 10 min at 180× *g* at 25 °C. The obtained ruminal epithelium cell pellets were resuspended in Dulbecco’s modified eagle medium (DMEM) and analyzed for cell viability using trypan blue. Finally, the cell density was adjusted to 1 × 10^6^ cells/mL, and the cells were seeded into 6-well cell culture plates (2 mL per well), 96-well cell culture plates (0.1 mL per well), and 24-well cell culture plates (1 mL per well) and incubated at 37 °C in 5% CO_2_ in a humidified incubator (Thermo Fisher Scientific, Waltham, MA, USA). The medium was replaced every 2 days. The animal experimental protocol adopted in this study was approved by the Ethics Committee on the Use and Care of Animals at Northwest A&F University (Yangling, China) and was conducted in accordance with the university’s guidelines for animal research (Approval No. 2021049).

### 4.2. Identification of Primary Bovine Rumen Epithelial Cells

The cells were grown at 50–60% confluency on the coverslips and then fixed with 4% paraformaldehyde, washed thrice with PBS, permeabilized with 0.02% Triton X-100, and blocked for 40 min with bovine serum albumin. The coverslips were washed thrice and incubated with primary antibodies specific for CK18 (BOSTER, Wuhan, China) and E-cadherin (Abways, Shanghai, China) overnight at 4 °C. Following incubation, the coverslips were probed with suitable secondary antibodies for 4 h at 25 °C, washed thrice with PBS, and strained with 4′,6-diamidino-2-phenylindole (DAPI, Sigma, St. Louis, MO, USA) nuclear stain. The coverslips were rinsed again, sealed with an anti-fluorescence quencher, and photographed using a fluorescence microscope (Nikon, Ni-U, Nagasaki, Japan).

### 4.3. LDH Activities Analysis

Membrane integrity was assessed by LDH activity [54]. After the cells were seeded into 24-well plates, the medium was collected daily for 7 consecutive days. The absorbance was measured at 450 nm wavelength according to the manufacturer’s protocol (Nanjing Jiancheng, Nanjing, China), and the LDH activity was calculated according to the following equation: LDH activity (U/L) = (OD_experiment_ − OD_control_)/(OD_standard_ − OD_blank_) × standard sample concentration × 1000.

### 4.4. Cell Viability Analysis

Primary bovine rumen epithelial cells were cultured up to 80% confluency, and the medium was replaced with fresh medium containing different doses of Sr (0, 0.1, 0.5, 1, 5, 10, 20, 50, 100, and 300 mmol/L) for 24 h at 37 °C in a 5% CO_2_ atmosphere. Each well was then treated with Cell Counting Kit-8 (CCK-8, ZETA LIFE, Menlo Park, CA, USA) at 37 °C in a 5% CO_2_ atmosphere for 4 h, and the absorbance was measured at 450 nm. The cell inhibition was calculated according to the following formula: cell inhibition = 1 − [(OD_drug_ − OD_blank_/OD_control_ − OD_blank_) × 100%]. The IC50 value was calculated by using the GraphPad Prism 8.4.3.

### 4.5. Cell Cycle Analysis

After treatment with Sr, bovine rumen epithelial cells were harvested with 0.25% trypsin-EDTA and fixed with 70% ethyl alcohol overnight at −20 °C; then, the cells were stained with 0.5 mL of propidium iodide/RNase staining buffer (BD Pharmingen™, Franklin Lakes, NJ, USA) for 15 min at 25 °C. Stained cells were immediately analyzed for propidium iodide fluorescence using flow cytometry (Coulter-XL). Cell cycle analysis was performed using the Cell Cycle platform in ModFit 3.0, and the PI was calculated according to the following formula: PI = (S + G2/M)/(S + G2/M + G0/G1) [55].

### 4.6. Intracellular Ca^2+^ Analysis

After treatment with Sr, bovine rumen epithelial cells were washed thrice with PBS and incubated with Fluo-4 AM (2 μm, Beyotime, Shanghai, China) for 4 h at 37 °C; then, the cells were incubated for 4 h at 37 °C after being washed thrice. Finally, fluorescence was measured on a multimode microplate reader (Tecan Spark, Männedorf, Switzerland).

### 4.7. RNA Extraction and RNA-Sequencing

Total RNA was extracted using the TRIzol reagent kit (Invitrogen, Carlsbad, CA, USA) from bovine rumen epithelial cells according to the manufacturer’s protocol. RNA quality was assessed on an Agilent 2100 Bioanalyzer (Agilent Technologies, Palo Alto, CA, USA) and checked using an RNase-free agarose gel. Total RNA was used to prepare a separate Poly-A isolated, strand-specific cDNA Library using NEBNext Ultra RNA Library Prep Kit (NEB #7530, New England Biolabs, Ipswich, MA, USA). The prepared cDNA was purified with AMPure XP Beads (1.0×) and sequenced on Illumina Novaseq6000 by Gene Denovo Biotechnology Co., (Guangzhou, China). The raw Illumina sequencing data were archived in the National Centre for Biotechnology Information-Sequence Read Archive (NCBI SRA, https://www.ncbi.nlm.nih.gov/sra accessed on 30 October 2022) under the accession number SUB12106664.

### 4.8. Protein Extraction and DIA Labelling

Total proteins were extracted from bovine rumen epithelial cells by the cold acetone treatment method, and protein quality was determined by the bicinchoninic acid (BCA) protein assay kit and examined using sodium dodecyl sulfate-polyacrylamide gel electrophoresis (SDS-PAGE). The proteins extracted from cells were reduced by dithiothreitol at 55 °C for 1 h, alkylated by iodoacetamide in the dark at 37 °C for 1 h, and digested to peptides at 37 °C for 16 h. The peptide mixture was re-dissolved in solvent A (A: 0.1% formic acid in water) and analyzed by on-line nano spray liquid chromatography-tandem mass spectrometry (LC-MS/MS) on an Orbitrap Fusion Lumos coupled to EASY-nLC 1200 system (Thermo Fisher Scientific, Waltham, MA, USA); the peptide sample was loaded onto an analytical column (Acclaim PepMap C18, 75 μm × 25 cm) with a 120 min gradient from 5% to 35% B (B: 0.1% formic acid in ACN). Finally, the column flow rate was maintained at 200 nL/min at a column temperature of 40 °C, and the electrospray voltage of 2 kV versus the inlet of the mass spectrometer was used. The mass spectrometer was run under a data-independent acquisition mode and automatically switched between MS and MS/MS mode. These experimental procedures and data analysis were performed by Gene Denovo Biotechnology Co., (Guangzhou, China).

### 4.9. Identification of Ca^2+^ Metabolism-Related Targets

With hypocalcemia as the keywords, Ca^2+^ metabolism-related targets were obtained from the GeneCards database (https://www.genecards.org/ accessed on 20 May 2022), Online Mendelian Inheritance in Man (OMIM) database (https://omim.org/ accessed on 20 May 2022), Therapeutic Target Database (TTD) (http://db.idrblab.net/ttd/ accessed on 20 May 2022), and DisGeNET database (https://www.disgenet.org/ accessed on 20 May 2022). The core targets of Ca^2+^ metabolism were obtained from transcriptome, proteome, and network pharmacology analyses.

### 4.10. Real-Time Polymerase Chain Reaction (RT-PCR)

Total RNA was extracted from Sr-treated cells using TRIzol reagent, and its concentration and purity were measured by an ultra-micro ultraviolet spectrophotometer (NanoDrop one, Thermo, Waltham, MA, USA). Then, cDNA was synthesized from total RNA by using reverse transcriptase. RT-PCR was performed with SYBR^®^ Premix Ex Taq™ (Perfect Real Time) Kit using CFX Connect Real-Time PCR System (Bio-Rad, Hercules, CA, USA). All primers were designed to span an exon-exon junction to avoid genomic DNA. The primers information is shown in Table 1. The 2^−ΔΔCT^ method was used to analyze the relative expression levels of genes.

### 4.11. Statistical Analysis

Differential expression for gene and protein obtained: clean data (clean reads) were obtained by fastp (version 0.18.0), and aligned with the reference genome mapped to the Bos taurus reference genome (Ensembl_release104) using HISAT2. 2.4 software (http://www.ccb.jhu.edu/software/hisat/ accessed on 15 August 2021). The mapped reads of each sample were assembled using StringTie v1.3.1 (https://ccb.jhu.edu/software/stringtie/ accessed on 15 August 2021) in a reference-based approach. For each transcription region, a fragment per kilobase of transcript per million mapped reads (FPKM) value was calculated to quantify its expression abundance and variations, using RSEM software [56]. RNA differential expression analysis was performed with DESeq2 (version 1.40. 1) software between the 0 mmol/L (Sr-0) and 20 mmol/L (Sr-20) Sr treatment groups. DEGs were identified with a false discovery rate (FDR, *q*-value) < 0.05 and log_2_ fold change (FC) (|log_2_FC| > 1). Statistically significant DEGs were illustrated using volcano plot analysis and visualized by hierarchical clustering analysis.

Raw DIA-MS data were processed and analyzed by Spectronaut X software (Biognosys AG, Switzerland) with default parameters. The retention time prediction type was set to dynamic iRT. The protein was qualitatively analyzed using a 1% *q*-value (FDR) cutoff on precursor and protein levels, and quantitative analyzed using the 1% *q*-value cutoff on the average top three filtered peptides. DEPs were analyzed by the Student’s *t*-test and Benjamini–Hochberg (BH), according to an absolute FC > 1.5 and *q*-value < 0.05 (Student’s *t*-test) as the screening criteria. The R package was used to generate Venn, heatmaps, and hierarchically clustered differential proteins based on normalized values.

GO and KEGG pathway enrichment analyses: gene function enrichment analysis (http://www.omicshare.com/ accessed on 23 August 2021) was performed using GO and Kyoto Encyclopedia of KEGG by the cluster Profiler R package, and *q*-value < 0.05 was used to indicate significantly enriched GO functions and KEGG pathways.

Enrichment analysis of DEPs was based on KEGG and GO databases (http://www.omicshare.com/ accessed on 10 September 2021), and a *q*-value < 0.05 indicated significantly enriched GO functions and KEGG pathways. The Rich factor in gene function enrichment analysis was calculated according to the number of genes in each category divided by the total number of genes in the category.

A PPI was generated using STRING (https://cn.string-db.org/ accessed on 10 September 2022) and Cytoscape software (version 3.8.2) to present the core and the biological interaction of hub genes.

Quantitative data analysis: statistical analyses were performed using one-way analysis of variance (ANOVA). The Shapiro–Wilk test was used for the normality test. All data are presented as the mean ± standard error of means (SEM). *p* < 0.05 was considered statistically significant. Each experiment was independently repeated at least thrice.

## 5. Conclusions

This study found the underlying targets of Sr-mediated Ca^2+^ metabolism regulation, namely, AHCYL2, SEMA3A, PTHLH, TGF-β2, and CYP11A1. These results will improve the current understanding of the regulatory effect of Sr on Ca^2+^ metabolism and provide a theoretical basis for Sr application in bovine hypocalcemia.

## Figures and Tables

**Figure 1 ijms-24-09383-f001:**
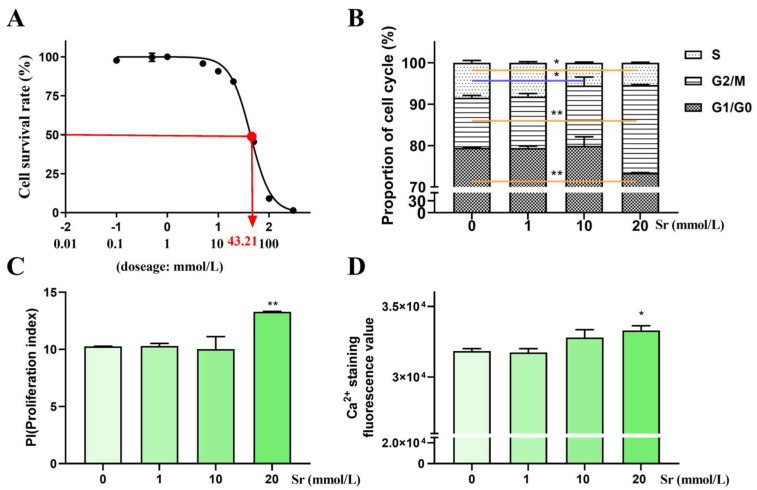
The analyses of IC50, cell cycle, and intracellular Ca^2+^ level treatment with Sr. (**A**) After 24 h of treatment with Sr, the IC50 of rumen epithelial cells was 43.21 mmol/L. (**B**) Percentages of the cell population distributing in the G0/G1, S, and G2/M phases. (**C**) Comparison of the proliferation index in different groups. (**D**) The fluorescence value of Ca^2+^ staining was significantly increased in the 20 mmol/L Sr treatment group, and showed an increasing trend with an increase Sr dose in bovine rumen epithelial cells; * *p* < 0.05, ** *p* < 0.01 as compared to the 0 mmol/L Sr treatment group.

**Figure 2 ijms-24-09383-f002:**
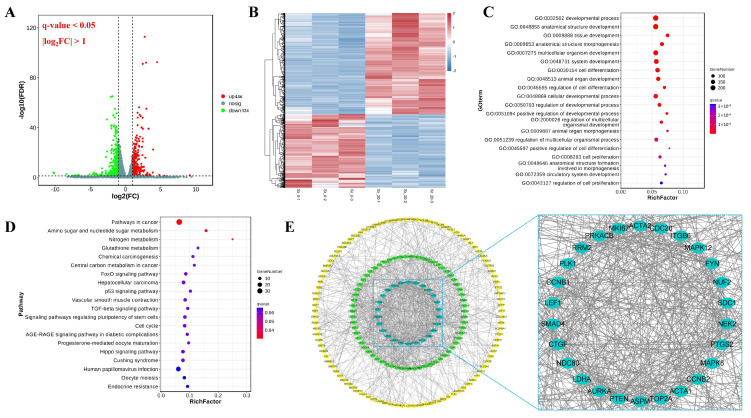
Transcriptomics profile comparison between control group (Sr-0) and Sr treatment group (Sr-20). (**A**) Volcano plot of the fold change and statistical significance. (**B**) Heatmap showed the changes in the expression pattern of DEGs. (**C**) GO enrichment analysis of DEGs; the top 20 significant enrichment pathways were listed. (**D**) KEGG enrichment analysis of DEGs; the top 20 significant enrichment pathways were listed; (**E**) PPI analysis of DEGs.

**Figure 3 ijms-24-09383-f003:**
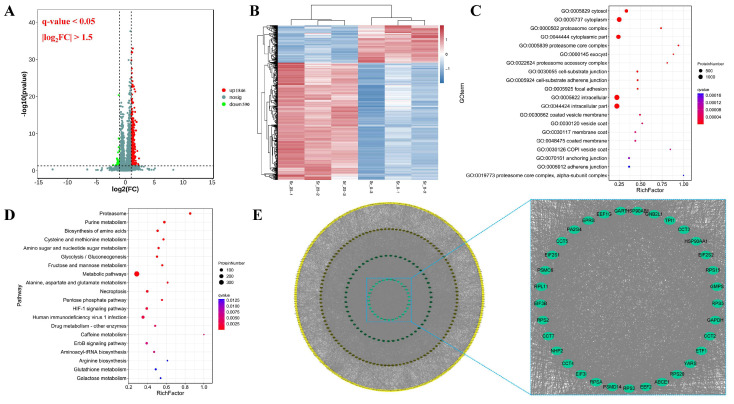
Proteomics profile comparison between the control group (Sr-0) and Sr treatment group (Sr-20). (**A**) Volcano plot of the fold change and statistical significance. (**B**) Heatmap showed the changes expression patterns of DEPs. (**C**) GO enrichment analysis of DEPs. The top 20 significantly enriched pathways were listed. (**D**) KEGG enrichment analysis of DEPs. The top 20 significantly enriched pathways were listed. (**E**) PPI analysis of DEPs.

**Figure 4 ijms-24-09383-f004:**
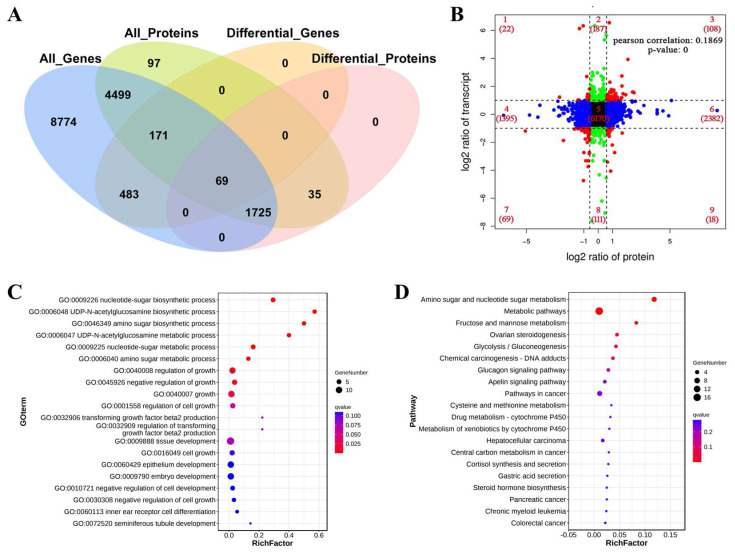
Associations analysis of transcriptomics and proteomics profiles. (**A**) Venn diagram of all mRNAs and proteins. (**B**) In the nine-quadrant diagram, the differential expressed genes and differential expressed proteins were screened according to the threshold of transcriptomics and proteomics. (**C**) GO enrichment analysis for DEPs-DEGs; the top 20 significantly enriched pathways were listed. (**D**) KEGG pathway analysis for DEPs-DEGs; the top 20 significantly enriched pathways were listed.

**Figure 5 ijms-24-09383-f005:**
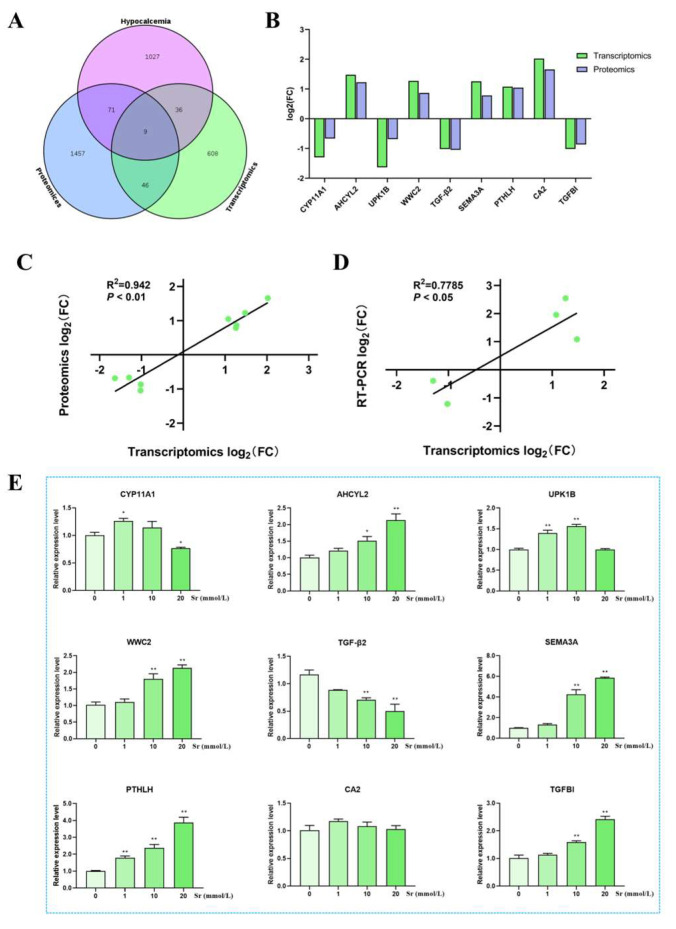
Screening and validation of Ca^2+^ metabolism and Sr core targets. (**A**) Venn diagram of hypocalcemia, transcriptomics, and proteomics profiles. (**B**) Core targets between hypocalcemia and Sr expression levels in transcriptomics and proteomics profiles. (**C**) Correlation analysis of core targets between transcriptomics and proteomics profiles. (**D**) Correlation analysis of CYP11A1, AHCYL2, SEMA3A, PTHLH, and TGF-β2 between RT-PCR and transcriptomics. (**E**) The core targets were verified using RT-PCR, * *p* < 0.05, ** *p* < 0.01, as compared to the 0 mmol/L Sr treatment group.

**Table 1 ijms-24-09383-t001:** Primers used for RT-PCR.

Gene Symbol ^a^	Accession No.	Product Size	Primer Sequence (5′ → 3′)
CYP11A1	NM_176644.2	219 bp	F: CTTGGAGGGACCATGTAGCC R: GCAATGTCATGGATGTCGTGT
AHCYL2	XM_005205707.4	279 bp	F: GCACAGTCAAGAAGATCCAATTTGC R: GTGCTGGCATTTCTTGCTCA
UPK1B	NM_174482.2	181 bp	F: GAGGAGAGGGTAAGCTTGGG R: TGGCTTCAAGCAGTGGGTAG
WWC2	XM_024986463.1	190 bp	F: CGCCCGGTTCCCCTATG R: GCTTGGTCAACCTGTCCC
TGF-β2	NM_ 001113252.1	264 bp	F: TCATGCGCAAGAGGATCGAG R: GCGGGATGGCATTTTCCGAG
SEMA3A	NM_001276701.2	224 bp	F: TCTTCCGAACTCTTGGGCAC R: GCCCCCAAAGTCATTCTTGC
CA2	NM_178572.2	201 bp	F: TCGCGGAGAATGGTCAACAA R: GTGAACCAGGTGTAGCTCGG
TGFBI	NM_001205402.1	273 bp	F: GAGCTCTGTGAAACTAGCCCC R: TGGGCTAACCGCCATGTTTA
PTHLH	NM_174753.1	132 bp	F: GGTTATTATTTCGGAGGAGGCG R: CTCTCGCTCTGGGGACTTAT
GAPDH	NM_001034034	117 bp	F: CCTGCCAAGTATGATGAGAT R: AGTGTCGCTGTTGAAGTC
18S	NR_036642.1	130 bp	F: ACCCATTCGAACGTCTGCCCTATT R: TCCTTGGATGTGGTAGCCGTTTCT

^a^ CYP11A1, Cholesterol side-chain cleavage enzyme; AHCYL2, adenosylhomocysteinase such as 2; UPK1B, uroplakin 1B; WWC2, WW, and C2 domain containing 2; TGF-β2, Transforming growth factor β2; SEMA3A, Semaphoring 3A; CA2, Carbonic anhydrase II; TGFBI, transforming growth factor beta-induced; PTHLH, parathyroid hormone-related protein.

## Data Availability

The datasets presented in this study can be found in online repositories. The names of the National Centre for Biotechnology Information-Sequence Read Archive (NCBI SRA) can be found below: https://www.ncbi.nlm.nih.gov/sra (accessed on 30 October 2022), SUB12106664.

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
