# Peer review of "A Network Pharmacology and Multi-Omics Combination Approach to Reveal the Effect of Strontium on Ca2+ Metabolism in Bovine Rumen Epithelial Cells"

_ijms, 2023, doi:10.3390/ijms24119383_

Round 1

Reviewer 1 Report

The study by Tan et al addresses the effects of Strontium (Sr) levels on Ca2+ metabolism in bovine rumen epithelial cells. This effect has not been previously investigated and it can be useful to set the theoretical basis for a potentially application of Sr in bovine hypocalcaemia.

They analyse the changes in the transcriptome and the proteome profiles of bovine rumen epithelial cells with cells treated with 20 mmol/L Sr compared to the untreated. Then they compare the DEGs with the DEPs and conclude that there is not much correlation, indicating that Sr treatment may affect post-transcriptional modifications.

This reviewer thinks that the study is overall interesting and well performed. The omic- approaches provide very interesting datasets and some of the potential targets have been validated by qPCR. In this regard, I think the analyse of these core targets at the protein level by Western Blot may add value to the report.  

In addition, I consider that, although the results are interesting, they are presented in a very descriptive way. Moreover, the discussion looks more like a results section. I consider results should be properly discussed, going into a deeper explanation of the findings and their functional relevance.  

Some other general comments about the figures and the text are:

·         The GO terms described on the text do not reflect what is on figure 1C.

·         Labels on figures 2, 3 and 4 are not readable. They should be made bigger.

·         Figure S1: E-cadherin staining is not clear. Is it not expected to find E-cadherin located at the cell membrane?

·         Some abbreviations should be defined the first time they are referred to (e.g. DEPs, DEGs, PPI and some others).

·        In the results section, a brief introduction of the experimental procedures should be added, so it is possible to understand what has been done. For example, that LDH activity is performed to check membrane integrity.  

Reviewer 2 Report

General comments

In this manuscript, the authors reported their study on the effect of strontium on Ca2+ metabolism in bovine rumen epithelia using RNA-sequencing and proteomics approaches. The methods are sound, the manuscript and the results are interesting and reasonably well presented. Their study provided useful information for understanding of Sr-mediated metabolic regulation and Sr application.

 Special comments

1.           In this study, the authors used primary epithelial cells isolated from bovine rumen.  In the manuscript, the authors did not give details of the number of animals used for the cell cultures in each experiment. Unlike the cell lines which are engineered to stabilise the important signalling pathways, those in the primary cells are not stable and there are differences among the individuals. Therefore, the study needs to use the cells isolated from an appropriate number of different animals to gain reliable results. I suggest that the authors clarify this the manuscript.

2.           Many simple sentences were used in the manuscript, especially in Discussion. Please change them to academic writing.

3.           In the manuscript, the symbols of gene and protein need to comply with the nomenclature roles. For example, gene symbol for cattle is in italic format. For more detail, see (https://en.wikipedia.org/wiki/Gene_nomenclature).

4.           Abstract. Please give brief description of animals and the number of animals used in the experiment in the abstract.

5.           Line 14. Change “explore” to “investigate”.

6.           Full name should be given when using the abbreviation at the first time.

7.           Figure 1A. The method for calculating IC50 was not given in the section of Materials and Methods (MM). If you used a model (such as inhibitory model or sigmoid model), this should generate more parameters, such as Emax and E0. These should be included in the results section.

8.           Figure 2A. Please label the values of the cut off criteria (FC and P (q-value) on the volcano plot.

9.           Figure 2C. Please explain the meaning of “Richfactor” and its calculation method in MM.

10.        Figure 4B. I suggest that the DEGs and DEPs in the interesting (or important) quadrants be listed in the supplementary files.

11.        Line 182. Rephrase.

12.        Line 185. Between which was there a positive correlation?

13.        Lines 186-187. What did “The normality test results of ….were not significantly changed” mean? Did you mean the test was not significant?

14.        Lines 205-207. Other mechanisms may also be involved.

15.        Line 127. Change “;” to “.”.

16.        Lines 219-221. Rephrase.

17.        Line 228. Change “correlated” to “were correlated”.

18.        Line 241. Change “,” to “.”.

19.        Line 250. Change “changes” to “differences”.

20.        Materials and Methods. Number of animals used in each study should be given.

21.        Statistical analysis. The providers of all software used in this study should be given, such as providers’ names, addresses or their websites.

22.        Statistical analysis. Suggest that you split this section into several sections, such as differential expression for gene and protein, KEGG and GO analyses, and statistical analysis (for data in Figure 1).

23.        Line 385. Please give the name and version of the reference genome.

24.        Line 390. Change “by” to “with”.

25.        Lines 393-394. DEGs cannot be generated with volcano plot. Volcano plot is done based on the differential analysis and cut off criteria.

26.        Line 409. Please give the database name and version for KEGG and GO analysis.

27.        Line 416. “Each experiment was independently repeated at least thrice”. Was this done with the cells from different animals or from a same animal.

Many simple sentences were used in the manuscript, especially in Discussion. Please change them to academic writing.

Reviewer 3 Report

In the abstract section Authors should indicate the statistical analysis applied in the study. Moreover, Authors should better emphasize, in the last sentence of abstract section, the significance of the study.

I suggest to avoid personal form (i.e. our, we, etc.) throughout the manuscript.

In the statistical analysis paragraph, Authors wrote the normal distribution of data has been assessed. Please indicate whether data passed the Normality test together with P value.

Results and Discussion sections are clear and well written and the findings have been justified with appropriate references.  

I suggest to add a separated conclusion section in which Authors better summarize the main findings gathered in their investigation as well as they should emphasize the significance of the study.

I strongly suggest to improve the quality of figures.

The English language should be improved throughout the manuscript

Round 2

Reviewer 2 Report

In this revised manuscript, the authors incorporated most of my comments and addressed some of my concerns. Its quality is now improved. I have the following minor comments:

1. Lines 398. Change “screened” to “illustrated”.

2. Line 416. Change “by the number of all differential genes…” to “by the number of all genes…”.

3. Line 419. Keep the paragraph structure consistent.

Some of the sentences need editing to improve the quality of English language writing. 
